# Using Co-Production to Develop “Sit Less at Work” Interventions in a Range of Organisations

**DOI:** 10.3390/ijerph18157751

**Published:** 2021-07-22

**Authors:** Kelly Mackenzie, Elizabeth Such, Paul Norman, Elizabeth Goyder

**Affiliations:** 1School of Health and Related Research, University of Sheffield, Regent Street, Sheffield S1 4DA, UK; e.such@sheffield.ac.uk (E.S.); e.goyder@sheffield.ac.uk (E.G.); 2Department of Psychology, University of Sheffield, Cathedral Court, 1 Vicar Lane, Sheffield S1 2LT, UK; p.norman@sheffield.ac.uk

**Keywords:** sitting time, sedentary, occupational, workplace, intervention development, co-production

## Abstract

Prolonged periods of sitting are associated with negative health outcomes, so the increase in sedentary jobs is a public health concern. Evaluation of interventions to reduce workplace sitting have suggested that participatory approaches may be more effective. This paper describes the use of co-production in four diverse organisations. Workshops with staff in each organisation were conducted to develop an organisation-specific strategy. The first workshop involved creative activities to encourage participants to develop innovative suggestions. The second workshop then developed a feasible and acceptable action plan. An ecological approach was used to consider behaviour change determinants at a range of different levels including intrapersonal, interpersonal, organisational, and environmental-level factors. 41 staff volunteered for workshops (seven in a small business, 16 in a charity, 15 in a local authority, and three in a large corporation). Of those, 27 were able to attend the first workshops and 16 were able to attend the second. Whilst there were some similarities across organisations, the smaller organisations developed a more tailored and innovative strategy than large organisations where there were more barriers to change and a more diverse workforce. Co-production resulted in bespoke interventions, tailored for different organisational contexts, maximising their potential feasibility and acceptability.

## 1. Introduction

Prolonged periods of sitting are associated with a range of negative health consequences including an increased risk of cardiovascular disease [1,2], type 2 diabetes [2,3], metabolic syndrome [4,5], colon cancer [6] and depression [7]. In addition, high levels of sitting time are associated with an increased risk of all-cause mortality [2,3,8,9,10]. There are several domains where prolonged sitting time could occur, but the workplace is a particular concern for several reasons. Due to the rise of the information economy [11], occupations largely composed of sitting and office-based work, e.g., administration and customer services, have increased [12]. Furthermore, adults spend approximately 60% of their waking hours in the workplace [13,14] and an observational study suggests that office workers in England spend 63% of their total daily sitting time, sitting at work [15]. It is this recognition of the increasing prevalence of sedentary occupations and the contribution the workplace makes to the health risks associated with prolonged sitting that has highlighted the need for interventions to reduce workplace sitting time [16]. Such studies have since emerged [17,18,19,20,21,22], but systematic reviews investigating intervention effectiveness have shown inconsistent results in terms of effect sizes and/or statistical significance of the findings [23,24,25]. 

A recent qualitative systematic review and proposed operational framework (see Appendix A) [26] identified the importance of using participatory approaches when developing interventions to reduce workplace sitting time. There is a spectrum of “user” participation from passive involvement, e.g., via education or coercion, to seeing users as partners in the co-production of an intervention. Co-production is being increasingly seen as a necessity for the development of complex public health interventions [27]. Developing interventions in collaboration with end users takes account of important contextual factors (e.g., culture, preferences, resources), producing interventions that are feasible, acceptable and sustainable, and have an increased chance of producing outputs that are translatable in the real-world, helping to address the ‘implementation gap’ [27,28]. In addition, co-production of interventions can create a sense of ownership to those supporting the development and ensure that the intervention content meets the needs of the end users [29]. Recent studies that have looked at co-production in relation to sedentary behaviour [30] and workplace health interventions [31] have reported several benefits of this approach, such as feelings of ownership and perceived ability to contribute resulting in a positive experience for participants [30], and increased acceptability and uptake of the intervention [31]. Whilst co-production is increasingly advocated for public health interventions [27,32], the empirical evidence relates mainly to intervention feasibility, acceptability, and uptake [27,28,31]. Guidance for good practice suggests that this increased uptake will increase the probability of intervention effectiveness [27,32], but, due to the complex social nature of co-production, there is no high-quality evidence for causality.

Complex intervention development processes and co-production techniques are rarely reported in detail [32]. However, in order to understand “what”, “why”, “how” and “when” decisions were made, and thereby ensure that intervention development processes can be replicated with fidelity, this detail is essential [33]. This paper therefore aims to provide a detailed report regarding the co-production of “Sit Less at Work” interventions in four organisations. Involving a range of organisations allowed the co-production methods used to be fully tested. The work presented in this paper is part of a larger project involving the development, implementation and evaluation of interventions to reduce workplace sitting time in a range of organisations (see full report [34] and earlier published papers [26,35] for further details). A linked paper also submitted to this Special Issue reports details of the process evaluation [36]. 

## 2. Materials and Methods

The intervention development process was conducted between March and June 2018. The operational framework (see Appendix A) [26] was used to guide the intervention development process. The operational framework identified the need to use a theoretical model as part of the intervention development [26]. The present research drew on the ecological model of sedentary behaviour [37]. This model provides a behaviour-specific prompt to focus on the domains within which relevant contextual factors (environmental, social, organisational, individual) influence certain sedentary behaviours. Ecological approaches have been advocated to address health behaviour change in research and practice [38] and have been used by other researchers when developing interventions to support staff to sit less at work [21,39,40,41]. The use of an ecological approach alongside co-production helped to ensure key characteristics were incorporated into the interventions, including tailoring to the needs of staff and the organisation, providing a menu of strategies and targeting multiple levels of influence.

Four organisations in South Yorkshire, UK, were purposefully recruited to reflect a range of sizes and sectors. The organisations included a small business (private sector); a charity (voluntary sector and large); a local authority (public sector and large); and a large corporation (private sector). A more detailed description of the participating organisations has been published elsewhere [35], and a summary can be found in Appendix A. For each participating organisation, senior management approval was obtained and a single named contact was provided. 

Convenience samples of desk-based employees (including staff, middle managers, and senior leaders) from each of the four participating organisations were recruited via an email sent by the named contact in each organisation. Convenience sampling methods were used, as the aim of this research was to establish how the interventions could be developed in “real world” settings. One implication of convenience sampling was that participants were drawn from the same population as the earlier qualitative study [35]. Participation in the previous study did not preclude participation in this present study, therefore there may have been some overlap. No incentives were used to recruit participants and compensation for their time was not provided. However, all participants were advised that managers had approved their participation within working hours. All participants provided written informed consent. Ethical approval for this phase of the study was obtained from the School of Health and Related Research Ethics Committee at the University of Sheffield, UK (reference no. 012219).

The co-production of the “Sit Less at Work” interventions was achieved via two one-hour workshops per organisation, held during working hours in meeting rooms at each organisation in order to increase access to as many participants as possible. All participants (regardless of job role) attended the same workshop in their organisation. Both workshops had a set of activities for participants to complete, with a researcher present to guide the activities and respond to any questions. However, the co-production principle of equality (which emphasises that everyone (the researcher and participants) is equal, everyone has assets (e.g., skills, abilities, time), and no one is more important [32]) was highlighted to all participants at the start of each workshop. 

Workshop 1 aimed to brainstorm initial ideas to help staff to sit less at work and used creative thinking activities to elicit ideas that were not constrained by cost or practicalities. An ecological approach was used to consider behaviour change determinants at a range of different levels including intrapersonal- (e.g., psychological and behavioural factors), interpersonal- (e.g., social support and addressing social norms), organisational- (e.g., local policies in workplaces, management involvement), and environmental-level factors (e.g., the workplace-built environment and the surrounding natural environment) [38]. For all activities, ideas generated were captured by participants on sticky notes. The first activity involved asking participants to consider different perspectives on how to sit less at work, in line with an ecological approach [38], including behaviour changes for: “you as an individual; your team/colleagues; your employing organisation; and your workplace environment”. The second activity required participants in small groups to list characteristics of a random word (selected from a page of random words, taken from Thinkertoys—a handbook of creative-thinking techniques, p. 167 [42]) and then force links between these characteristics and potential sit less at work initiatives. This activity encouraged participants to think “outside of the box”. These activities were conducted with participants standing and moving to different corners of the room to model the idea of sitting less at work. 

Ideas gathered from Workshop 1 were then reviewed by the researcher who had attended the workshop and thematically analysed using the following pre-determined themes (which related back to the ecological approach): (1) What an individual could do; (2) What a team could do or how a team/colleagues could support an individual; (3) What the organisation could do; and (4) How the office environment could be changed. These ideas were then fed back to the participants from the first workshop to ensure that they were an accurate reflection of the discussions. After some further refinement based on this feedback, a final list of ideas, categorised by theme, was brought to the second workshop. 

Workshop 2 allowed time and space to explore those ideas that participants felt were most feasible and suited to their organisation, during a group discussion. In addition, specific barriers to implementation were identified by participants and ways to overcome them considered. Finally, plans for implementing each idea were explored by reviewing several questions, i.e., when should each idea be implemented during the intervention period, were any specific permissions required to action these ideas, who needed to be contacted, and who would take responsibility for these actions? All these data were captured using a bespoke intervention planning proforma (see Appendix A).

After the second workshop, information gathered was put into a detailed action plan for each organisation. This included: a description of each action which made up the “Sit Less at Work” intervention; the timing and frequency of each action during the 12-week intervention period; potential barriers to the actions and suggestions of how to overcome these; and who was to be responsible for implementing each action. An intervention summary document was also produced for each organisation, which identified the timings for each of the actions during the 12-week intervention period. Both documents were shared with the volunteers and their feedback incorporated into the final versions. 

At the end of each workshop, participants completed a brief questionnaire providing basic demographic and work-related information. 

## 3. Results

### 3.1. Intervention Development

A total of 41 volunteers from the four organisations (n = 7 in the small business, n = 16 in the charity, n = 15 in the local authority, and n = 3 in the large corporation) initially expressed interest in participating in the workshops. Of those, 27 were able to attend the first workshops (n = 6 in the small business, n = 10 in the charity, n = 9 in the local authority, and n = 2 in the large corporation) and 16 were able to attend the second workshops (n = 4 in the small business, n = 6 in the charity, n = 4 in the local authority, and n = 2 in the large corporation). Table 1 shows participant characteristics by organisation and by workshop and highlights that most participants were well-educated, White British women who worked full-time. However, there were some differences by organisation. For example, in the small business most participants were younger men, and in the local authority, the participants had a higher average age than the other organisations.

### 3.2. Intervention Content

Ideas elicited from participants in Workshop 1 presented by organisation and by level of influence are shown in Appendix A. Ideas ranged from those that were easily feasible, e.g., encouraging staff to drink more water and go for lunchtime walks, to those that were not feasible at all either due to financial constraints, e.g., the purchase of sit-stand desks, or due to practical or ethical limitations, e.g., getting an electric shock if sat for too long. 

The outputs from the second workshops are presented in a series of Appendix A. These tables describe the final “Sit Less at Work” interventions, including an intervention summary and detailed action plan for each organisation. Actions for each “Sit Less at Work” intervention were staggered over the 12-week period to ensure there were not too many during any one week. The following sub-sections will highlight the intervention plans for each organisation. 

#### 3.2.1. Small Business

There were two overarching actions that were intended to be on-going throughout the 12-week intervention period, which included the use of social media to promote the “Sit Less at Work” intervention externally, and to have the intervention as an agenda item for team meetings to support the development of a new health and wellbeing policy incorporating sitting less at work. Actions to be implemented in a stepwise manner included: weekly email communications from the Managing Director (MD) to encourage staff to sit less at work; using computer prompts to remind staff to take regular sitting breaks; using an exercise ball instead of a chair; setting up and participating in various competitions which involved movement such as press-up competitions, ping-pong and computer games; exercising with a set of office dumbbells; walking to the shops for lunch; and using wireless headsets to allow walking whilst on the phone.

Identified barriers to implementation primarily related to lack of time, potential workload pressures and impact on service delivery. However, there were some more practical barriers to implementation such as where to safely store the office dumbbells and concerns that some staff might feel excluded if they do not want to participate. 

#### 3.2.2. Charity

An initial communication was planned to be sent to managers and all staff informing them of the sit less intervention prior to the start of the intervention. Throughout the 12-weeks, it was intended that all team meetings would be held stood-up or have some standing/movement incorporated, and policies and guidelines would be amended to ensure sitting less at work was included. Actions to be implemented on a weekly basis during the intervention period included: regular email communications from the Chief Executive promoting the intervention; staff to develop personal daily step targets; staff to celebrate “sit less” stories over internal social media platform; staff to join the lunchtime walking and/or running groups; the implementation of a clear office/desk policy (i.e., the centralisation of office equipment and stationery).

Practical barriers to implementation were identified and included difficulties of taking notes during standing meetings, and the fact that there was only one shower which might stop staff participating in the lunchtime running group. Broader barriers included lack of time, organisational culture and concerns that some staff may not want to or be able to participate. 

#### 3.2.3. Local Authority

There were three overarching actions to continue throughout the 12-week period, which included: holding team meetings stood-up or incorporating standing or moving into team meetings; highlighting the issue of sitting less at work in one-to-one meetings; and integrating sitting less and moving more at work into workplace guidelines. Weekly actions included: regular communications via the Chief Executive’s blog encouraging staff to sit less at work; setting up team step competitions; having regular team standing breaks; and having a weekly message over the Tannoy system reminding everyone to stand or move.

Barriers relating to implementation were quite broad and referred to lack of time, workload pressures, concerns that middle management may not be supportive, and health and safety concerns. Practical barriers related to the need for Occupational Health or Human Resources approval to include sitting less and moving more into workplace guidance and one-to-one meetings and the fact that not all staff would have access to appropriate technology such as a smart phone or wearable device (e.g., a FitBit) to take part in team step competitions. 

#### 3.2.4. Large Corporation

Actions spread across the 12-week period included: regular emails from the Area Manager with “sit less” suggestions; weekly “sit less” challenges; and incorporating movement into the monthly team meetings. The barriers to implementation related to staff engagement and workload pressures. 

### 3.3. Implementation Plans

#### 3.3.1. Small Business

As most staff from the small business, including the MD, were involved in the development of the intervention, determining implementation plans and deciding who was to action these plans was clear and straightforward. This allowed named individuals to be allocated different tasks to support the implementation of the intervention.

#### 3.3.2. Charity

A member of staff in the Human Resources team, Tom (pseudonym), whose role included responsibility for the health and wellbeing of staff, was identified by the workshop participants as a key individual to support the implementation of the intervention. A briefing meeting was held to inform Tom of the aims of the intervention and what actions the staff had decided upon. Tom then became the main point of contact and took on the responsibility for implementing the intervention actions.

#### 3.3.3. Local Authority

A briefing paper summarising the intervention plans was presented by the Director of Public Health at a Directors meeting to formally approve the intervention. An implementation planning meeting was then held. At this meeting, a member of the Human Resources team was given responsibility for implementation. It should be noted that at this stage the “message over Tannoy” action was vetoed by the Human Resources team due to concerns for health and safety. Therefore, this action was removed from the final intervention plan. 

#### 3.3.4. Large Corporation

The two volunteers who attended the workshops were keen to ensure the implementation of the “Sit Less at Work” intervention. However, they did not have the capacity to implement it alone. To gain support for the intervention from colleagues, they discussed it at their team meeting on two separate occasions. However, both times there was “no appetite” for implementing the intervention amongst their colleagues. This organisation, therefore, decided to opt out of the project at this stage and no further implementation plans were made. 

## 4. Discussion

Four “Sit Less at Work” interventions were co-produced and implementation plans made with volunteer staff from each organisation using an appropriate operational framework to guide the process [26].

### 4.1. Intervention Content

The four interventions had several notable similarities. First, planned communications about the intervention were to come from a senior leader in the organisation to provide explicit “permission” for staff to participate. This was in line with previous studies, which used co-production when developing interventions [20,39] and maximised an enabler, suggesting that management support could help encourage staff to sit less at work [35]. Second, incorporating sitting less and moving more into health and wellbeing policies or guidelines was suggested by participants from the small business, charity, and local authority, with the aim of ensuring that staff felt permitted to sit less. This action has been recommended as a means of supporting a shift in organisational culture [43], an identified barrier to sitting less at work [35]. Third, participants from the larger organisations (charity, local authority, and large corporation) produced similar ideas, which were also consistent with previous studies that used participatory approaches [39,44]. These ideas included: incorporating standing or moving into team meetings [39,44]; replacing periods of sitting with activity, e.g., holding standing or walking meetings [39]; or encouraging lunchtime walks or some other form of exercise [44]. Finally, monitoring step counts was also suggested by participants from all the larger organisations.

Although it is reassuring that multiple studies using participatory approaches have produced similar intervention strategies, it is not clear whether these strategies were themselves effective. A review of behaviour change strategies used in sedentary behaviour reduction interventions by Gardner et al. [45], identified self-monitoring, problem solving, and restructuring the social or physical environment as the “most promising” techniques to support behaviour change. For interventions based in the workplace, the review also found that “very promising” interventions were associated with a primary aim of targeting sedentary behaviour (as opposed to physical activity or other health promotion initiatives). The four interventions presented in this paper had the primary aim of encouraging and supporting staff to sit less at work. Furthermore, some of the actions suggested by workshop participants were in line with self-monitoring, e.g., developing personal targets for daily step counts, and restructuring the social environment, e.g., having competitions, celebrating “sit less” success stories, and having team standing breaks. Problem solving was incorporated during the development of the interventions by addressing barriers to implementation as part of Workshop 2. Other studies that have used participatory approaches [20,39,44,46] also used some, but not all, of the “very promising” behaviour change techniques identified by Gardner et al. [45] but demonstrated mixed effectiveness. It is possible that it is the process of developing and implementing interventions, in addition to their content or the use of specific behaviour change techniques, that determines their success [26].

There were some notable differences in the four “Sit Less at Work” interventions. In particular, participants from the small business produced suggestions that were more innovative, compared to the other organisations, such as: installing software onto their computers to remind them to regularly get up and move; incorporating a range of competition elements into the intervention, e.g., ping pong, press-ups, and computer games, which incorporated movement; using an exercise ball to sit on rather than a standard chair; doing some gym-style exercises with a set of office dumbbells; setting up a rota for staff to take it in turns to walk to get lunch for the team; purchasing and using wireless headsets to allow staff to move around when on the phone; and using social media to promote their positive health and wellbeing actions. Except for the use of computer prompts [39,44,47], these actions were also quite different to interventions in previous studies.

The reasons the small business was so distinct in terms of the intervention content could be related to the size and sector of this organisation and/or the gender and age of the staff. Being such a small organisation in the private sector meant that it was simple to develop and plan the implementation of innovative actions and purchase small amounts of low-cost equipment without a lot of bureaucracy, which has been identified as a barrier for larger organisations [48,49]. In addition, as most of the staff involved in the development process were young men, this led to the inclusion of more “male-oriented” actions, which encompassed a level of competitiveness [50,51], e.g., press-up competitions and the use of computer games. The larger organisations suggested more conservative ideas, e.g., step competitions, incorporating standing into meetings. Participants from these larger organisations had a clear understanding of some of the broader barriers to implementation, such as the bureaucracy related to health and safety and equality and diversity policies. In contrast, there was more flexibility in the small business, as structures relating to such policies were less rigid.

### 4.2. Implementation Plans

There were no formal plans for implementation prior to commencing this study. It was anticipated that implementation would be an iterative and pragmatic process, dependent on the needs of each organisation. Furthermore, planning such interventions too far in advance could have resulted in difficulties particularly in terms of staff attrition and turnover. Barriers to implementation of each of the initiatives in the four interventions were explored during the workshops. Many of the reported barriers were operational or practical such as where to store the office dumbbells in the small business or how to take notes during standing meetings, as reported by participants from the charity. However, some of the barriers were broader and more ingrained in the culture of the organisations, such as lack of time, workload pressures and concerns about lack of support from middle managers. These broader barriers were also identified in a study which explored existing barriers and enablers regarding sitting less at work [35]. Plans for the implementation of the “Sit Less at Work” interventions (which included addressing the broader, organisational cultural barriers) were supported by engaging management and gaining their commitment for the intervention. Engaging management has been highlighted as important by other studies [21,39,40,52,53] and was also identified as a key step in the operational framework [26] as an enabler to sitting less at work [35]. Senior management approval for the project was sought and obtained from all participating organisations. However, only the MD from the small business was involved in all stages of the development process.

For the charity and local authority, the implementation process was more complex. Although approval for the project had been obtained from the organisations’ Chief Executives, due to capacity commitments, they were unable to be directly involved in the implementation plans. Therefore, they had to delegate this responsibility to appropriate members of staff. This was consistent with a previous study where managers from relevant departments were used to facilitate the logistics of implementing an intervention [53]. The reason the implementation process was more complex in the charity and local authority could have been due to the larger sizes of these organisations. As organisation size increases, so does organisational complexity and formalisation [54], increasing the levels of bureaucracy and making it harder for leaders to achieve the desired levels of commitment [49].

The large corporation dropped out at the point of planning implementation. As no further formal data were collected on the reasons for dropping out, it was difficult to fully understand why the staff were no longer willing to participate. However, there were two possible explanations. First, although team leaders had approved the project, they had immediately delegated the responsibility for participation to more junior staff who had no capacity for implementing the intervention. Second, there were only two volunteers involved in the intervention development workshops. If there had been more interest during the development phase, then the ideas put forward might have been more representative of all staff and hence increased the likelihood of wider engagement.

### 4.3. Potential Benefits and Limitations of Co-Production

Using co-production allowed for the development of bespoke “Sit Less at Work” interventions that were tailored to the needs of staff in each organisation and maximised their potential feasibility and acceptability. Co-production also helped to ensure that the interventions were appropriate for the different organisational contexts. Furthermore, co-production is known to encourage “buy-in” from other members of staff as there is a sense of ownership, given that the intervention is developed by colleagues/peers [29] rather than “outsiders” (researchers) who may not be perceived to have the insights required to develop an intervention suitable for the organisation. In addition, co-production can provide the support required to change policy and/or practice in the real-world by fostering context-specific decision-making and supporting knowledge translation [28].

There are several noteworthy limitations to co-production. First, co-producing an intervention can be time-consuming, as, by its very nature, it is an iterative process and it is not clear at the outset how long the process will take to complete [29]. This was not a particular issue in the current research, as staff volunteers were recruited easily, but is certainly an issue to consider when planning to adopt co-production techniques. Second, the iterative nature of co-production and the fact that participants can feel empowered through the process of co-production, can mean the intervention may change over time. Therefore, it can be difficult to know when the intervention is finalised, which can be a threat to intervention fidelity [29]. Finally, due to the nature of co-production, participants’ ideas may not fit within an a priori behavioural theory and hence cannot be easily categorised and shown to be in line with such theories. This could explain why the ideas presented in this paper only touched upon some of the “very promising” behaviour change techniques identified by Gardner et al. [45].

Co-production has been used in other studies to develop interventions in order to reduce workplace sitting time [39,40,44,55]. However, it is unclear if the use of co-production results in more effective interventions [26]. Taking a “top-down”, researcher-led, evidence-based approach to intervention development, is likely to be less time consuming and have limited threats to intervention fidelity and internal validity. However, this mechanism lacks creativity and the ability to tailor to the needs of staff and/or an organisation, which is important when considering translating this process into different organisational contexts and maximising external validity.

### 4.4. Stengths and Limitations of the Study

Strengths of this study included the use of a theoretical model to support intervention development. The use of a theoretical model is in line with the existing evidence-base [32] and ensured that theoretically sound interventions were developed which considered a range of behaviour determinants across multiple levels of influence. Furthermore, the use of co-production helped to ensure interventions were developed which addressed the needs of staff in each organisation and helped to enhance the potential acceptability and feasibility of the developed interventions.

There were also some limitations to this study. First, the small number of volunteers and the fact the volunteers were obtained using convenience sampling techniques was likely to have impacted the representativeness of the volunteers. Due to data protection concerns, data relating to the demographic make-up of each organisation were not available, so it was not possible to establish how representative our samples were. Second, there were high dropout rates from those who initially volunteered to those who participated in the first workshop (66%), and between the first and second workshops (59%), which also could have impacted the representativeness of the volunteers. It is possible that those who were particularly interested in the research topic (including those who had been involved in the earlier qualitative study [35]) volunteered to participate. However, this is likely to reflect engagement in “real world” co-produced intervention development. It is also possible that the low levels of participation were linked to participant availability, rather than lack of interest. This may also occur in the “real world”. Of particular note was the lack of initial engagement and then the subsequent participation of only two employees from the large corporation which is likely not to have been representative, and contributed to the organisation deciding to opt out of the implementation stage of the project.

## 5. Conclusions

This paper describes the co-production of four “Sit Less at Work” interventions. It provides a detailed description of not only what was developed, but also why, how, and when decisions were made, and explores plans for implementation. The use of co-production techniques and an operational framework to guide the discussions in Workshop 1 ensured that the interventions developed in Workshop 2 encompassed a range of strategies that targeted multiple levels of influence and were tailored to the needs of the staff in each organisation. Co-production resulted in bespoke interventions, tailored for different organisational contexts, maximising their potential feasibility and acceptability. These interventions have since been rolled out and a process evaluation undertaken, details of which have been reported in a linked paper published in this Special Issue [36].

## Figures and Tables

**Table 1 ijerph-18-07751-t001:** Participant characteristics for workshops 1 and 2.

Characteristic	Small Business	Charity	Local Authority	Large Corporation	Total
Total number of employees	8	488	4146	119,300 *	-
Workshop 1					
Total number of participants	6	10	9	2	27
Number of management-level participants	3	2	2	2	9
Mean age (years)	32	37	46	42	39
Women	1	7	7	2	17
Ethnicity					
- White British	6	8	8	2	24
- Other	0	2	1	0	3
Highest educational attainment					
- Degree or equivalent	1	5	7	2	15
- Higher education	2	3	0	0	5
- A level or equivalent (13 school years)	2	0	2	0	4
- GCSEs grade A*-C or equivalent (11 school years)	1	2	0	0	3
Full-time	5	7	8	2	22
Workshop 2					
Total number of participants	4	6	4	2	16
Number of management-level participants	3	0	2	2	7
Mean age (years)	37	30	50	42	40
Women	1	5	3	2	11
Ethnicity					
- White British	4	6	4	2	16
- Other	0	0	0	0	0
Highest educational attainment					
- Degree or equivalent	1	4	3	2	10
- Higher education	1	0	1	0	2
- A level or equivalent (13 school years)	2	1	0	0	3
- GCSEs grade A *-C or equivalent (11 school years)	0	1	0	0	1
Full-time	3	5	3	2	13

* Although there were 119,300 employees in total in the large corporation, the recruitment email only reached 25 members of staff who were based within the branch that had agreed to participate in the project.

## Data Availability

The datasets used and/or analysed during the current study are available from the corresponding author on reasonable request or can be found in the full PhD project report at https://etheses.whiterose.ac.uk/28889/ (accessed on 6 June 2021).

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
