# Peer review of "Using Co-Production to Develop “Sit Less at Work” Interventions in a Range of Organisations"

_ijerph, 2021, doi:10.3390/ijerph18157751_

Round 1

Reviewer 1 Report

Mackenzie and co-authors reported on strategies explored to reduce sedentary time at work in four different organisations. The authors conducted two workshops with employees from four different organisations to brainstorm possible ideas to reduce sitting time at work. This study forms part of a larger study aimed at reducing sitting time at work.

The study was well-written, however, some clarification and suggestions below may improve the quality of the manuscript.

The first major concern is the drop out rate of the participants in the workshops. Only 27 of the 41 participants attended the first workshop, while only 16 participants attended the second workshop. This amounts to a greater than 60% drop-out rate. Also in the large corporation, only 2 individuals of the ~120 000 employees participated. Therefore, it is unlikely that this is a representative sample.  The lack of participation and engagement from the employees in the large corporation is evidence from the results of the study.  

Second, the article lacks empirical evidence about the approach followed to arrive at the intervention, and secondly whether the interventions implemented had any effects on sitting time. Hence, making the claims about the successes/failures of co-production is not supported by any evidence. Furthermore, the authors propose a “hybrid model” but it is not clear what this suggestion is based on, as there is no new evidence produced in this study to support this.  Although the author briefly mentions behaviour change theory, the theoretical framework is not adequately explored. Combining the process of the workshops and the outcome of the intervention will strengthen the findings.

Reviewer 2 Report

This article was an interesting study focused on developing interventions to reduce sedentary time in collaboration with end users. Overall, I found the article to be well written, and they chose a very important topic to study given the health risks associated with prolonged sitting. However, additional detail on the co-production methods and implementation is needed.

  1. Please clarify if participants who participated in workshops were different that those who completed focus groups in ref. 34. It stated the organizations are the same as the prior publication.
  2. Were barriers identified in workshops different than prior focus groups?
  3. More detail about workshop process would improve this manuscript. Who led workshops? Were participants compensated for participation? How were resulting themes analyzed?
  4. What happened during the implementation stage? Were any metrics collected to assess implementation success? Did it reduce sedentary time? Would be strengthened by any results during implementation.
  5. For those involved in workshops, what were their perceptions of interventions at work?
  6. How similar are the companies who participated vs those who didn’t? Of the employees who participated, how similar are they to demographics of other employees within the company?
  7. Are these intervention plans specific to that one company or can they be translated to other companies of similar size? The recommendations seem fairly broad and it’s unclear why the action plan/implementation plan wouldn’t translate to other companies of other sizes and types?

Reviewer 3 Report

The manuscript is relevant and describes an interesting intervention approach. There are minor points in the attached file.

Round 2

Reviewer 2 Report

n/a